# Two types of North American droughts related to different atmospheric circulation patterns.

Angela-Maria Burgdorf [1,2], Stefan Brönnimann [1,2], Jörg Franke [1,2]

[1]Oeschger Centre for Climate Change Research, University of Bern, Bern, 3012, Switzerland
[2]Institute of Geography, University of Bern, Bern, 3012, Switzerland

*Correspondence to*: Angela-Maria Burgdorf (angela-maria.burgdorf@giub.unibe.ch)

# Two types of North American droughts related to different atmospheric circulation patterns.

Angela-Maria Burgdorf [1,2], Stefan Brönnimann [1,2], Jörg Franke [1,2]

[1]Oeschger Centre for Climate Change Research, University of Bern, Bern, 3012, Switzerland
[2]Institute of Geography, University of Bern, Bern, 3012, Switzerland

*Correspondence to*: Angela-Maria Burgdorf (angela-maria.burgdorf@giub.unibe.ch)

**Abstract.** Proxy-based studies suggest that the southwestern USA is affected by two types of summer drought, often termed "Dust Bowl"-type droughts and 1950s-type droughts. The spatial drought patterns of the two types are distinct. It has been suggested that they are related to different circulation characteristics, but lack of observation-based data has precluded further studies. In this paper, we analyze multi-annual summer droughts in North America since 1600 in tree-ring based drought reconstructions and in a global, monthly 3-dimensional reconstruction of the atmosphere. Using cluster analysis of drought indices, we confirm the two main drought types and find a similar catalog of events as previous studies. These two main types of droughts are then analyzed with respect to 2-meter temperatures (T2m), sea-level pressure (SLP), and 500 hPa geopotential height (GPH) in summer. 1950s-type droughts are related to a stronger wave-train over the Pacific-North American sector than "Dust Bowl"-type droughts, whereas the latter shows the imprint of a poleward shifted jet and establishment of a Great Plains ridge. The 500 hPa GPH patterns of the two types differ significantly not only over the contiguous United States and Canada but also over the North Atlantic and the Pacific. "Dust Bowl"-type droughts are associated with positive GPH anomalies, while 1950s-type droughts exhibit strong negative GPH anomalies. In comparison with 1950s-type droughts, the "Dust Bowl"-type droughts are characterized by higher SSTs in the North Atlantic. Results suggest that atmospheric circulation and SST characteristics not only over the Pacific but also over the extratropical North Atlantic affect the spatial pattern of North American droughts.

## 1 Introduction

Since the turn of the 21[st] century, prolonged drought events have afflicted large parts of North America, predominantly the southwestern United States (hereafter Southwest) (Seager, 2007; Weiss et al., 2009; Cayan et al., 2010; Seager and Vecchi, 2010; Seager and Hoerling, 2014). In recent years, exceptionally severe droughts struck California (Griffin and Anchukaitis, 2014; Seager et al., 2014a, 2015), the Great Plains (Hoerling et al., 2012, 2014; Livneh and Hoerling, 2016) as well as the Texas-Northern Mexican region (Seager et al., 2014b). However, drought conditions

are a regular feature of the climate in the western United States and have repeatedly affected the region in the past (*e.g.,* Cook et al., 2007; Seager et al., 2009). A particularly strong multi-annual drought event in the instrumental record period was the decadal-scale "Dust Bowl" drought, which coincided with the Great Depression and had tremendous economic and social effects (Worster, 1979). Proxy-data provide evidence that even longer and more severe drought periods, so-called "megadroughts", have occurred in the past (*e.g.,* Woodhouse and Overpeck, 1998; Cook et al., 2007, 2010, 2016; Stahle et al., 2007). Severe droughts in the paleoclimate record include decadal to multidecadal droughts during medieval times (~AD 900-1300), characterized by not only persistent aridity but also increased temperatures over western North America (Woodhouse and Overpeck, 1998; Cook et al., 2004; Woodhouse et al., 2010).

In light of greenhouse gas-induced global warming, this temperature-drought relationship (with the concurrence of increasing aridity and rising temperatures) is alarming and climate model simulations suggest a possible increase in drought frequency and duration in the 21$^{st}$ century (Seager et al., 2007; Cayan et al., 2010; Seager and Vecchi, 2010; Dai, 2013; Cook et al., 2014a, 2015, 2018a; Ault et al., 2016). In order to be able to cope with the challenges associated with the projected increase in aridity and thus drought risk in the future, it is important to better understand the dynamics behind multi-annual drought events in the western United States. Since severe multi-annual droughts are limited in the era of observation-based climate data, analyses have to be extended into the reconstruction era. Reconstructions based on proxy data, however, are mostly restricted to interpretations as surface temperature and moisture. Here we analyze past drought periods in 3-dimensional reconstruction that is based on data assimilation.

During the last decade, considerable progress was achieved in isolating the mechanisms behind multi-annual droughts in the western U.S.. Both proxy-based studies (Woodhouse and Overpeck, 1998; McCabe et al., 2004; Routson et al., 2016) and model simulations suggest that oceanic forcing by both the Pacific and to lesser degree the Atlantic acts as a trigger (*e.g.,* Schubert et al., 2004a; b, 2009; Seager, 2007; Cook et al., 2008; Kushnir et al., 2010; Seager et al., 2015; Baek et al., 2019). In particular, a cool Pacific and a warm Atlantic, especially in their tropical regions, are conducive to droughts such as the 1930s "Dust Bowl" drought, demonstrating a combined impact of both ocean basins (McCabe et al., 2004; Schubert et al., 2004b, 2009; Feng et al., 2008; Kushnir et al., 2010). However, the roles of individual ocean basins remain less clear. For the period of instrumental SST observations after 1856, a persistent La Niña-type SST pattern during all prolonged droughts is found as well as a cold Indian Ocean during most of them (Seager et al., 2005; Herweijer et al., 2006; Seager, 2007). The contribution of the Atlantic Ocean is shown to be only minor. In contrast, Nigam et al. (2011) find a key role of the Atlantic SST in their observation-based analysis of 20th century drought and wet periods in the Great Plains. A significant influence of North Atlantic SSTs variations on multi-decadal droughts in the continental U.S. is also found by Enfield et al. (2001). While the general setting of a cool Pacific and warm Atlantic is overall sufficient to produce drought conditions in AMIP-type model simulations, strong droughts require further mechanisms such as a land-surface feedback or the effect of dust (see Schubert et al., 2004b; Cook et al., 2008, 2009, 2013, 2014b). Furthermore, studies have shown that droughts are strongly influenced

by internal atmospheric variability of the climate system unrelated to oceanic forcings or a combination thereof (*e.g.,* Hoerling et al., 2009; Seager and Hoerling, 2014; Cook et al., 2018b; Baek et al., 2019).

Considering this complex interplay of oceanic forcings, internal atmospheric variability as well as feedback mechanisms, it is apparent that not all North American droughts are alike. Fye et al. (2003) analyzed tree-ring based multi-annual drought reconstructions and found two different types of drought based on their spatial pattern: "Dust Bowl"-type droughts and 1950s-type droughts. These distinct spatial patterns possibly suggest different underlying dynamics of the atmospheric circulation leading to prolonged drought events. Woodhouse et al. (2009) analyzed two types of drought with respect to associated 500 hPa geopotential height fields and found evidence for both tropical and extratropical influences. Their study was based on severe single-year droughts and restricted to the 1949-2003 period. Further, they focused on atmospheric circulation during December through March to represent conditions of the cold season. Analyzing a large sample of droughts with respect to upper-level atmospheric circulation was not hitherto possible and thus only droughts in the instrumental record period have so far been analyzed, mostly focusing on the "Dust Bowl" drought. Hemispheric and global upper-level fields were reconstructed by Griesser et al. (2010) back to 1881 and used to study upper-level circulation during the "Dust Bowl" drought (Brönnimann et al., 2009, 2012), but not a larger set of droughts. The "Twentieth Century Reanalysis" (Compo et al., 2011) now extends back to 1851 and was used to study, *e.g.,* the heat waves associated with the "Dust Bowl" drought (Cowan et al., 2017) or the Atlantic imprint on droughts in the USA (Nigam et al., 2011).

Here we follow up on this work and further analyze the two types of multi-annual drought suggested by Fye et al. (2003) in a new monthly global 3-dimensional reconstruction back to 1600 (Franke et al., 2017), which allows focusing on atmospheric circulation. We analyze the imprint of the two types of drought in different fields and find that they differ significantly in their imprint over the contiguous United States and the extratropical North Atlantic.

The paper is organized as follows. Section 2 describes the data sets used (reconstructions and reanalyses) as well as the methods. Analyses of droughts during the past 400 years are presented in Section 3. Additionally, we analyze the most recent drought, namely 2000-2015, that was excluded from previous statistical analyses. In Section 4 we discuss the results in the light of possible oceanic forcing. Finally, conclusions are drawn in Section 5.

## 2 Data and Methods

### 2.1 Data sets used

Droughts in our study are addressed using the Living Blended Drought Atlas version 1 (LBDAv1) (Cook et al., 2010). This data set is an updated version of the North American Drought Atlas (NADA) (Cook et al., 1996, 1999, 2004) and provides annual values of the summer (June-August, JJA) Palmer Drought Severity index (PDSI) for the past two millennia based on tree ring reconstructions. The reconstruction includes 1845 tree ring chronologies and covers the North American continent at a spatial resolution of 0.5°x0.5° degree latitude. This allows for a regionally better

characterization of drought variability compared to the NADA. The LBDAv1 is highly consistent with the global hydroclimate reconstruction (PHYDA) that reconstructs the PDSI at ~2° resolution, based on a multi-proxy approach (Steiger et al., 2018). The comparison of PDSI fields from the LBDAv1 and PHYDA confirmed the very good agreement of the two PDSI products. Because no significant differences exist between the two data sets over the southwestern U.S., we decided to use the LBDA due to its higher resolution and in order to stay consistent with the work of Fye et al. (2003).

For comparison with Fye et al. (2003) and due to the availability of the reconstruction data set EKF400 (see below) we restricted the analysis to the post-1600 period for PDSI and atmospheric circulation. In order to include droughts in the 21$^{st}$ century we used the Palmer Modified Drought index (PMDI) from the Living Blended Drought Atlas version 2 (LBDAv2) (Cook et al., 2010). While the LBDAv1 ends in 2005, the LBDAv2 is updated until 2017. It is based on the LBDAv1 (Cook et al., 2010) and calculates a PMDI by recalibrating the PDSI using GHCN 5km instrumental data and a Kernel Density Distribution Method. The LBDAv2 is furthermore limited to the contiguous U.S. compared to North America in the LBDAv1 which is why version 2 is only used for the 21$^{st}$ century drought. This latest drought, ca. 2000-2015, is too recent to be statistically analyzed according to our definitions (see below) but is briefly addressed as a separate event. To ensure a consistent drought detection metric we had to scale the LBDAv2 (PMDI) to the LBDAv1(PDSI) (see below).

To address fields of atmospheric circulation we analyzed 500 hPa geopotential height (GPH), sea-level pressure (SLP), and air temperature 2 meters above ground (T2m) as well as circulation indices from the EKF400 data set (Franke et al., 2017). T2m and sea surface temperature (SST) from the model input are almost identical which justifies the use of T2m also over the ocean. EKF400 is a global, monthly, three-dimensional reconstruction based on an off-line assimilation of early instrumental data, documentary data, and proxies (tree ring width, late wood density) into an initial condition ensemble of 30 global model simulations using an Ensemble Kalman Filter technique. The data set is given at a resolution of 4° and covers the period 1603-2004. The model is constrained, among other forcings, with annual sea-surface temperature reconstructions from Mann et al. (2009), to which we have added intra-annual, ENSO-related variability (see Bhend et al. (2012), for a method description). In this study, we analyze the ensemble mean. While the PHYDA data set offers reconstructed global T2m fields for the boreal growing season June through August (JJA) at a ~2° resolution it does not include further atmospheric variables. We therefore limit our analysis of the atmospheric circulation contributing to long-term summer droughts to the EKF400 reconstruction. However, we offer a comparison of surface temperature composites for the four most recent drought periods in EKF400, PHYDA and furthermore in Berkeley Earth (Rohde et al., 2013a, 2013b), HadCRUT4.6 (Morice et al., 2012) as well as 20CRv2c (Compo et al., 2011) in the supplement (Supplement Figure S7). The reconstructions are generally in good agreement with the observations, but some differences remain.

In order to analyze drought dynamics of droughts in the 21$^{st}$ century we use the ERA-Interim reanalysis at a 2° lat x 2° spatial resolution (Dee et al., 2011), which is updated to the present.

The circulation indices analyzed are the Pacific North American (PNA) pattern (Wallace and Gutzler, 1981) and the latitude of the zonal mean subtropical jet over North America (i.e., the latitude of the maximum zonal mean zonal wind at 200 hPa as in Brönnimann et al. (2015), but restricted to 120° W to 60° W). For the PNA index, monthly anomaly time series were standardized based on the 1901-2000 reference period. All analyses are performed using the ensemble mean of 30 members as well as using the "Best Member", i.e., choosing for each warm season the member that best reproduces a global set of 34 high-quality tree-ring width chronologies (Brönnimann, 2015). The best member minimizes the cost function Eq. (1):

$$J(x) = (\mathbf{y} - H[\mathbf{x}])^T \mathbf{R}^{-1} (\mathbf{y} - H[\mathbf{x}]) , \tag{1}$$

with $\mathbf{y}$ denoting the tree ring width, $H[\mathbf{x}]$ is the tree ring width as modeled by the VS lite tree ring model (Breitenmoser et al., 2014) and $\mathbf{R}$ the corresponding error covariance matrix determined from using instrumental data in the 20[th] century (see Breitenmoser et al., 2014, for details). In contrast to the ensemble mean, whose variance decreases back in time, the best member has a stable variance over time.

Note that the SSTs used to drive the model that formed the basis of the assimilation exhibit suppressed interannual as compared to decadal variability (Franke et al., 2013). For instance, indices of El Niño and of the Pacific Decadal Oscillation (PDO) are very similar. Since the focus of this paper is on multi-annual drought, defined by means of a 5-yr filter (see below), the suppression of interannual variability does not markedly affect our results (given the similarity between El Niño and PDO indices in the SSTs underlying our assimilation, we refer to a "negative PDO/La Niña" in this paper).

## 2.2 Methods

Our starting point is the paper by Fye et al. (2003), which was based on a previous version of a PDSI reconstruction with a 2° lat × 3° spatial resolution, derived from 426 tree-ring chronologies (Cook et al., 1996, 1999). We first reproduced their analysis using the LBDAv1 (Cook et al., 2010) which has improved areal coverage and spatial resolution. The agreement between the two data sets was generally good, clear deviations were found for some of the drought periods, pointing to the need for re-classifying the drought events using LBDAv1.

All multi-annual droughts found in our preliminary analysis as well as documented by Fye et al. (2003) primarily affect the Great Plains and the Southwest. For that reason, we decided to subset our domain in order to even better capture the spatial signal of droughts in this particular drought-prone area. In doing so, our detection method revealed, in addition to the pervious drought catalog, further multi-annual drought periods that are very prominent in the relevant domain but remain disguised when looking at entire North America. We thus concentrated our drought classification on the region 22°-52°N and 130°-85°E, a domain that includes, beside the Great Plains and the Southwest, prominent drought regions in the Mississippi Valley, Northern Mexico, and the southern Canadian Great Plains. It however excludes the East Coast and the tropical South as well as Alaska and most of Canada.

For the definition of drought periods, we proceeded in the following way: First, an index for the surface area affected by drought (PDSI < -1) (following Fye et al., 2003) for every time step (summer) was calculated. The index was then filtered with a 5-yr moving average, as the focus is on multi-year droughts. Years with >33% of surface area affected by drought were then selected. These years are considered "drought years". Five or more drought years in succession are defined as drought periods. Single years with a smaller percentage than 33% in-between continuous drought years are included in the drought period. This resulted in a list of 17 drought periods for the period 1600-2005, which is displayed in Table 1 (see Supplementary Table S1).

In a next step, the spatial patterns of the drought periods were clustered. The clustering was based on the fields of time-averaged PDSI per drought period in LBDAv1. The individual drought periods were weighted with the square root of their length in years, furthermore, grid cells were area weighted. The sample size of 17 drought periods is quite sensitive to the cluster numbers, the chosen spatial and temporal domain as well as the clustering approach itself. Limiting the clustering to two clusters resulted in a robust classification of the droughts and is furthermore consistent with the literature (e.g., Fye et al., 2003) (see Supplementary Figure S1 and S2 for the PDSI pattern of the individual droughts). In terms of the clustering method, we chose a combined approach of ward hierarchical clustering and $k$-means clustering. Ward hierarchical clustering was used to determine the cluster centers, which were then used as a starting point for the $k$-means clustering (see Supplementary Figure S3 and S4). With this combined method, the resulting clustering affiliations for the droughts are identical to the ward hierarchical clustering, pointing to the robustness of the clustering.

The "turn-of-the-century" drought was excluded from the clustering analysis due to limited data availability (LBDAv1 only covers the years up to 2005, whereas drought conditions in the southwestern U.S. continued into the 2010s). Using LBDAv2, which ends 2017 (due to the different areal extent and spatial resolution, the dataset had to be fitted to the LBDA using regression), our definition detects a drought from 2000 to 2015. Interestingly, its spatial pattern correlates negatively with both drought types which indicates that it is potentially characteristic of an alternative drought type (Supplementary Figure S5). We therefore excluded the most recent drought, henceforth called 21[st] century drought, from the clustering and statistical analyses and instead analyzed it independently.

For the analysis of climate fields, each drought period was first expressed relative to a reference period that comprised 5 years before and 5 years after the drought period. Then we calculated composites for the types of drought events for different seasons. In the following, we focus on the summer-mean (JJA) fields of 500 hPa GPH, SLP, and T2m. We did compare the ±5 years composites approach to the simpler approach of using anomalies from the long-term mean, which resulted in qualitatively similar patterns for the case of SLP (Supplementary Figure S6). However, using a common climatology is not a good option for variables that have strong centennial trends such as T2m and GPH. For these variables, spurious signals may appear as our drought sample is small and the two types of droughts are not equally distributed over time, so they will be aliased by global warming trends.

The anomalies of the individual composites as well as their differences were tested using a non-parametric Wilcoxon-Mann-Whitney test.

# 3 Results

Plotting the first two principal components of the PDSI during the drought periods well separates two distinct clusters, explaining 23.3% and 17.2% of the total variance respectively (Supplementary Figure S4). A comparison with Fye et al. (2003) reveals, that our approach tends to depict more drought periods than theirs, but 11 (perhaps 12) out of our 16 periods can be attributed to corresponding periods in Fye et al. (2003), though the length differs (see Table S1). Among these 11 (12) droughts, eight (nine) were classified in the same cluster as in Fye et al. (2003) if we term our first cluster "Dust Bowl"-type droughts and our second cluster 1950s-type droughts (see Table S1). We therefore kept that nomenclature. Fye et al. (2003) use a different clustering method which could explain part of the difference in the classification. We find four (five) drought periods that are not classified by Fye et al. (2003) whereas they find two droughts that our analysis does not capture. Out of the 11 (12) drought periods, three are assigned to opposite clusters in our study compared to Fye et al. (2003): We found the 1772-1776 and the 1869-1874 droughts to be "Dust Bowl"-type droughts while Fye et al. (2003) depict them as 1950s-type droughts. In their study, however, the duration of these two droughts is double in length compared to ours, which could serve as an explanation as to the different classification. The third drought that is classified differently is the "Civil War Drought" of the 1850s and 1860s where the duration of the drought is coincident in both two studies apart from one year. It appears that the drought changes its character within the drought period from La Niña to El Niño condition which could possibly explain this discrepancy. There are three major differences between our study and Fye et al. (2003): First, Fye et al. (2003) use the first version of gridded summer PDSI reconstruction (Cook et al., 1996, 1999) which is a less sophisticated version of the LBDAv1. Second, we use a different drought detection metric and third, they use the entire domain over the contiguous U.S. whereas we focus on a subsetted drought-prone domain. Given the methodological differences between Fye et al. (2003) and our study, it is remarkable how similar both drought catalog and clustering results are, pointing to their robustness.

Figure 1 shows averaged PDSI values of two clustered drought types (a-c). There are distinct differences visible between the two types of drought. While both the "Dust Bowl"-type and 1950s-type affect the midwestern and southwestern United States, "Dust Bowl"-type droughts stretch far into the Pacific Northwest, whereas 1950s-type droughts stretch down into Mexico. The difference between the "Dust Bowl"-type and 1950s-type (9 and 7 droughts, respectively) shows that these two features are statistically significant (Fig.1 c). A clear NW-SE dipole arises.

In order to investigate potential differences in atmospheric circulation associated with the different spatial pattern of the two drought types, we performed a composite analysis in EKF400, namely in the 500 hPa GPH, SLP and T2m anomaly fields.

The results for boreal summer show that "Dust Bowl"-type and 1950s-type droughts are markedly different particularly at 500 hPa GPH (Fig. 2 a,c,e). Only the former exhibits a clear "Great Plains ridge" (see Namias, 1982) (a), whereas the 1950s-type drought displays negative anomalies over large parts of North America (c). The difference field shows a band of positive anomalies stretching from Alaska, the Pacific Northwest across the continent and the Atlantic towards Northern Europe and Scandinavia (e). Over Alaska, the Northwest, the Great Lake Region and over the northern Atlantic, the differences are statistically significant. Moreover, there are significant positive anomalies in the extratropical Pacific.

Both types of droughts have a less strong SLP imprint (Fig. 2 b,d,f). The signal resembles the 500 hPa GPH fields especially north of 50°N. Significant differences between the two drought types are located across Alaska and the central northern Pacific.

In order to address the circulation on a global scale, we also analyzed the position of the zonal mean subtropical jet at 200 hPa (maximum of zonal mean zonal wind, see Brönnimann et al., 2015) for the two drought types both in the ensemble mean and the best member (Fig. 3). "Dust Bowl"-type droughts are associated with anomalous northward shift of the jet over North America while during 1950s-type droughts the jet is shifted slightly further south relative to the preceding and following 5-year period. According to a heteroscedastic t-test, the differences in the jet position between the two drought types are significant ($p = 0.013$) for the ensemble mean (for the best member analysis, $p = 0.032$).

Summer droughts in the United States are often associated with precipitation deficits in winter and spring (*e.g.,* Weiss et al., 2009). While we performed our analysis for preceding winter (December-January-February) and spring (March-April-May) seasons as well, we do not include the results here. This is because, over North America, EKF400 is based on mostly tree ring proxy-data until the mid-19th century. Therefore, outside of the growing season, the reconstruction basically reflects an atmospheric model simulation.

The composites for T2m reveal that both types of events are clearly accompanied by a negative PDO/La Niña signature in the Pacific (Fig. 4), where cool surface temperatures along the western coast of North America in a horseshoe shaped pattern surround a core of warmer surface waters in the central North Pacific. However, larger differences appear over the contiguous USA as well as over the Atlantic. "Dust Bowl"-type droughts are 0.1-0.5 °C warmer over the USA and southern Canada than 1950s-type droughts (relative to their corresponding references), with largest differences in the Great Lakes region and the Pacific Northwest. Over central and northern Canada, on the other hand, 1950s-type droughts are markedly warmer (0.1 – 0.4°C) compared to "Dust Bowl"-type droughts. The temperature differences between the two drought events are not significant over the Pacific though the negative PDO is less pronounced for "Dust Bowl" cases. In contrast, the signal over the Atlantic differs between the two drought types. "Dust Bowl"-type droughts exhibit positive anomalies in the extratropical North Atlantic (the region of the North Atlantic current) and significantly warmer temperatures in the Barents Sea. Note that 5 years preceding and following

the drought periods were used as a reference. In the presence of slowly varying SSTs, this means that part of the signature of low-frequency modes such as the AMO might be missing (see discussion).

The latest multi-annual drought we include in the composite analyses is the 1950s drought. As mentioned before, the 21$^{st}$ century drought from 2000-2015 classified based on the LBDAv2 was excluded from previous analyses since its pattern correlated negatively with the two drought types.

Figure 5a shows the spatial pattern (PMDI) of the 21$^{st}$ century drought. The drought pattern is characterized by a pronounced dipole consisting of drought conditions in the West, especially the Southwest and wet conditions over the Midwest.

In 500 hPa GPH (Fig. 5c) and SLP (Fig. 5d) the 21st century drought exhibits strong positive anomalies in high latitudes and over Greenland and in contrast, strong negative anomalies over the northeastern Atlantic. These large-scale composite patterns differ strongly from the composites of the "Dust Bowl"-type and 1950s-type drought, nevertheless the dipole Greenland - northeastern Atlantic resembles the 1950s-type drought. The T2m composite (Fig. 5b) shows a negative PDO/La Niña signature and in the Pacific, however, also a warm equatorial Pacific e.g. an El Niño signature. In the far north, more precisely over Greenland and the Arctic Archipelago, the turn-of-the-century drought is characterized by a strong warming signal (0.2 – 1.5°C). While the GPH and SLP fields if anything, resemble the 1950s-type drought, the surface temperature field with above normal temperatures in the western U.S. corresponds with the "Dust Bowl"-type.

## 4 Discussion

Droughts in the USA have been shown to be closely linked to SST anomalies (*e.g.,* Hoerling, 2003; McCabe et al., 2004; Schubert et al., 2004b; a; Seager, 2007). Both types of drought exhibit generally negative SST anomalies over the tropical Pacific similar to a negative PDO/La Niña mode. This is consistent with the above-mentioned studies. However, not all La Niña-type (or negative PDO) events lead to drought, the pronounced 1972-1975 event did not, for instance (*e.g.,* Seager and Hoerling, 2014). Pu et al. (2016) argue that the drought development in La Niña years requires anomalous warming over the tropical North Atlantic in spring. Nigam et al. (2011) find that a positive phase of the AMO favors droughts in North America. In fact, their GPH pattern for positive AMO phases in summer is very similar to the anomalies found for the "Dust Bowl" drought (Brönnimann et al., 2012). There appear to be differences in Atlantic temperatures between the two drought types (Fig. 4), which could be related to modes of Atlantic multi-decadal variability such as the AMO or the NAO-coupled variability of the gyre circulation as discussed in the literature (i.e., warming in the Gulf Stream and Greenland, Iceland and Norwegian Seas (GIN) seas, but cooling in between; Curry and McCartney, 2001; Eden and Jung, 2001; Sun et al., 2015; Wills et al., 2019).

Differences between the two drought types appear most significantly in the 500 hPa GPH composites. The pattern for 1950s-type droughts exhibits a clear wave pattern, which however does not project well onto the PNA pattern, which is usuallyf defined for winter only. The pattern for "Dust Bowl"-type droughts is more zonally symmetric and the wave imprint is weaker. At the same time, a poleward shift of the subtropical jet is found. Upper-air observations and reconstructions for the "Dust Bowl" drought indicate a poleward-shifted jet over North America (Brönnimann et al., 2009) and the development of a "Great Plains ridge" (Namias, 1982). The composite of the "Dust Bowl"-type droughts corresponds well with these findings.

The wave-pattern in 500 hPa GPH over the North Pacific for 1950s-type droughts might be an indication that the relation to Pacific SSTs is stronger. Both types of drought are related to a negative PNA index in the preceding winter and spring period (not shown). There has been some debate as to what extent SST variability modes such as ENSO modify the mode itself or merely excite a "fixed" pattern (Straus and Shukla, 2002), but a negative PNA index is expected in La Niña winters. Skillful reconstruction of 500 hPa GPH in winter would be necessary to decide whether the difference between the two drought types only emerges in spring and summer or already in the preceding winter. The results of Woodhouse et al. (2009) suggest the latter.

The poleward shift of the jet, in the zonal average, has also been shown to be related to SSTs (Staten et al., 2018). Again, both Atlantic and Pacific might contribute. In particular, a negative PDO/La Niña like pattern over the Pacific has been shown to conduce to tropical expansion (Allen et al., 2014). Moreover, Atlantic SSTs were demonstrated to play a role in the form of a positive (negative) AMO for a poleward (equatorward) shift of the zonal mean jet (Brönnimann et al., 2015). It is thus not surprising that "Dust Bowl"-type droughts, with a warmer North Atlantic, have a more poleward shifted jet, although SST differences in the North Atlantic are not significant.

In all, this suggests that the two patterns of North American drought emerge from slightly different combinations of Atlantic and Pacific influences that operate via a Pacific wave train and a poleward shift of the jet, respectively. Atmospheric circulation and SSTs over both ocean basins, the Pacific and the North Atlantic, shape drought development in North America.

Seager (2007) analyzed the turn-of-the-century drought (1998-2004) and argued that it started with strong La Niña conditions (1998/99, which is not yet part of our drought definition due to the 5-yr moving average), that subsequently weakened and gave rise to a second phase (after 2002) with even slight El Niño conditions. Seager (2007) noted that the turn-of-the-century might not yet be over. It did indeed persist and was predominantly characterized by La Niña conditions interrupted by weak El Niño phases. It ultimately ended with the strong El Niño of 2015/16.

Our analysis shows that the 21st century drought exhibited a markedly different spatial pattern that can be attributed to neither the "Dust Bowl"- nor the 1950s-type drought. While drought conditions in the last two decades have been studied extensively, the focus has been on specific regions and exceptionally severe events like the 2011-2014 California Drought (Griffin and Anchukaitis, 2014; Seager et al., 2014a, 2015). Here we analyze the atmospheric circulation in summer (JJA) from 2000-2015 and find a distinct pattern in both the 500 hPa height and SLP fields.

Recent drought events often are found to be associated and amplified with anomalously warm temperatures (e.g., Griffin and Anchukaitis, 2014). The surface temperature fields show warmer than usual temperatures in the U.S. Southwest as well as a strong warming signal over Arctic Archipelago, Greenland and the northern Atlantic.

Here, we have demonstrated that both the spatial pattern and atmospheric circulation of the 21$^{st}$ century drought differ considerably compared to 16 multi-annual drought periods in the past five centuries. This highlights the likelihood of global warming contributing to the complex drought dynamics, not only amplifying drought duration and severity (Seager and Vecchi, 2010; Woodhouse et al., 2010; Cook et al., 2015, 2018a) but possibly changing the character of droughts in the future.

## 5 Conclusion

The new global 3-dimensional climate reconstruction EKF400 allows for the first time to study atmospheric circulation during a sufficiently large number of prolonged dry spells in a climate reconstruction. We find two distinct drought types over North America, that differ with respect to atmospheric circulation and SSTs. While the 1950s-type droughts exhibit a wave-train pattern over the Pacific North American sector, the "Dust Bowl"-type droughts show the imprint of a poleward shifted jet and a "Great Plains ridge". SSTs exhibit a negative PDO/La Niña-like pattern in the Pacific for both types, but slightly stronger for 1950s-type drought whereas the "Dust Bowl"-type droughts show a stronger warming of the Atlantic. Differences in SSTs (which are not significant) and differences in atmospheric circulation (which are significant) are consistent with each other and with the literature.

Results suggest that atmospheric circulation and SSTs characteristics over both the Pacific and the extratropical North Atlantic affect the spatial pattern of North American droughts, leading to two main drought types. Given the possible increase of droughts in a future climate, deepening our understanding of drought mechanisms in North America is important. Further refined reconstruction of past climate hydroclimate could help.

**Data sets**

**EKF400**

Franke, J., Brönnimann, S., Bhend, J. & Brugnara, Y. World Data Center for Climate at Deutsches Klimarechenzentrum. http://dx.doi.org/10.1594/WDCC/EKF400_v1 (2017).

**ERA-Interim**

Dee, D. P., Uppala, S. M., Simmons, A. J., Berrisford, P., Poli, P., Kobayashi, S., Andrae, U., Balmaseda, M. A., Balsamo, G., Bauer, P., Bechtold, P., Beljaars, A. C. M., van de Berg, L., Bidlot, J., Bormann, N., Delsol, C., Dragani, R., Fuentes, M., Geer, A. J., Haimberger, L., Healy, S. B., Hersbach, H., Hólm, E. V., Isaksen, L., Kållberg, P., Köhler,

M., Matricardi, M., McNally, A. P., Monge-Sanz, B. M., Morcrette, J.-J., Park, B.-K., Peubey, C., de Rosnay, P., Tavolato, C., Thépaut, J.-N. and Vitart, F.: The ERA-Interim reanalysis: configuration and performance of the data assimilation system, Q. J. R. Meteorol. Soc., 137(656), 553–597, doi:10.1002/qj.828, 2011.

**LBDAv1, LBDAv2**
Cook, E. R., Seager, R., Heim, R. R., Vose, R. S., Herweijer, C. and Woodhouse, C. A.: Megadroughts in North America: placing IPCC projections of hydroclimatic change in a long-term palaeoclimate context, J. Quat. Sci., 25(1), 48–61, doi:10.1002/jqs.1303, 2010.

**PHYDA**
Steiger, N. J., Smerdon, J. E., Cook, E. R. and Cook, B. I.: A reconstruction of global hydroclimate and dynamical variables over the Common Era, Sci. Data, 5, 1–15, doi:10.1038/sdata.2018.86, 2018.
Steiger, N.J. Zenodo https://doi.org/10.5281/zenodo.1154913 (2018).

**20CRv2c**
Compo, G. P., Whitaker, J. S., Sardeshmukh, P. D., Matsui, N., Allan, R. J., Yin, X., Gleason, B. E., Vose, R. S., Rutledge, G., Bessemoulin, P., Brönnimann, S., Brunet, M., Crouthamel, R. I., Grant, A. N., Groisman, P. Y., Jones, P. D., Kruk, M. C., Kruger, A. C., Marshall, G. J., Maugeri, M., Mok, H. Y., Nordli, Ø., Ross, T. F., Trigo, R. M., Wang, X. L., Woodruff, S. D. and Worley, S. J.: The Twentieth Century Reanalysis Project, Q. J. R. Meteorol. Soc., 137(654), 1–28, doi:10.1002/qj.776, 2011.

**Berkeley Earth**
Rohde, R., A. Muller, R., Jacobsen, R., Muller, E. and Wickham, C.: A New Estimate of the Average Earth Surface Land Temperature Spanning 1753 to 2011, Geoinformatics Geostatistics An Overv., 01(01), 1–7, doi:10.4172/2327-4581.1000101, 2013a.
Rohde, R., Muller, R., Jacobsen, R., Perlmutter, S. and Mosher, S.: Berkeley Earth Temperature Averaging Process, Geoinformatics Geostatistics An Overv., 01(02), 1–13, doi:10.4172/2327-4581.1000103, 2013b.

**HadCRUT4.6**
Morice, C. P., Kennedy, J. J., Rayner, N. A. and Jones, P. D.: Quantifying uncertainties in global and regional temperature change using an ensemble of observational estimates: The HadCRUT4 data set, J. Geophys. Res. Atmos., 117(8), D08101, doi:10.1029/2011JD017187, 2012.

**Code**

Clustering is performed with the following R packages: *hclust{stats}* and *kmeans{stats}*.

The R code for the subtropical jet position is published as supplement to Brönnimann et al. (2015).

**Author contribution**

SB had the initial idea for this paper. A-MB performed most of the analysis and figure designs with contribution of SB. JF provided the reanalysis data and assisted with interpreting the results. A-MB and SB drafted the manuscript in consultation with JF. All authors provided critical feedback and helped shape the manuscript.

**Competing interests**

There are no competing interests present.

**Acknowledgements**

This work was supported by the Swiss National Science Foundation (project 162668 RE-USE) and EU project 787574 (ERC Grant PALAEO-RA).

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

**Figures**

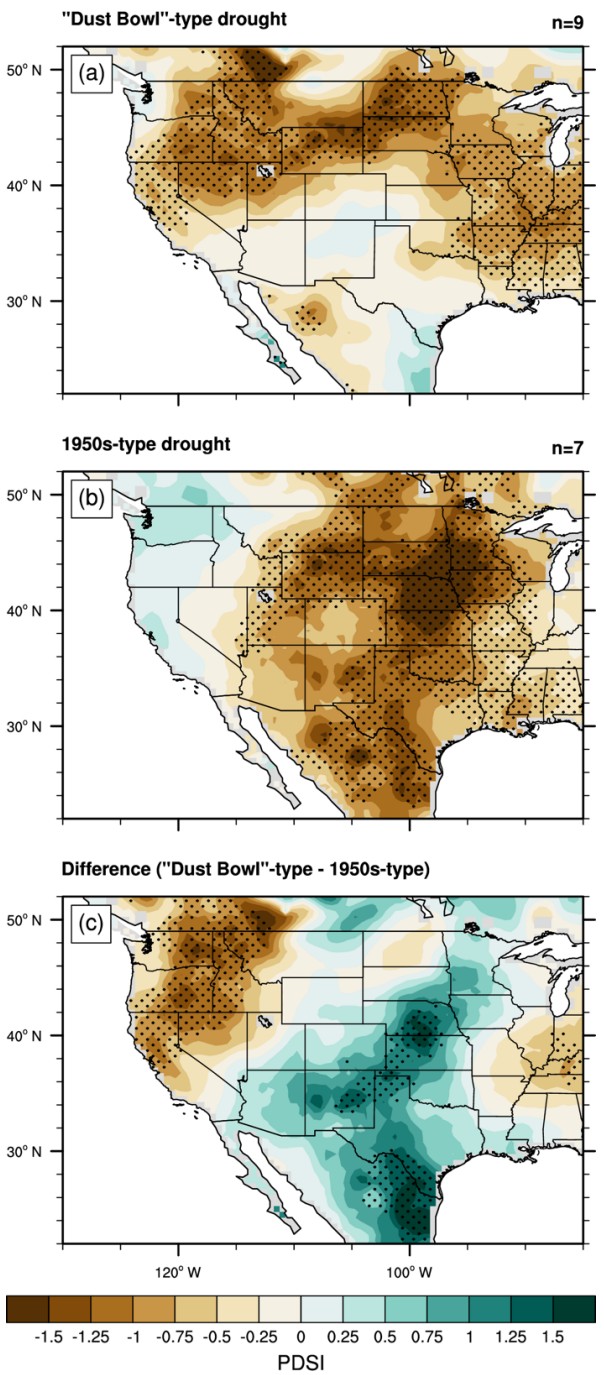

Figure 1: Averaged PDSI values from LBDAv1 for "Dust Bowl"-type droughts (a) and 1950s-type droughts (b) since 1600. The difference between "Dust Bowl"- and 1950s-type is shown in (c). Stippling indicates significance at the 95% level based on a non-parametric Wilcoxon-Mann-Whitney test.

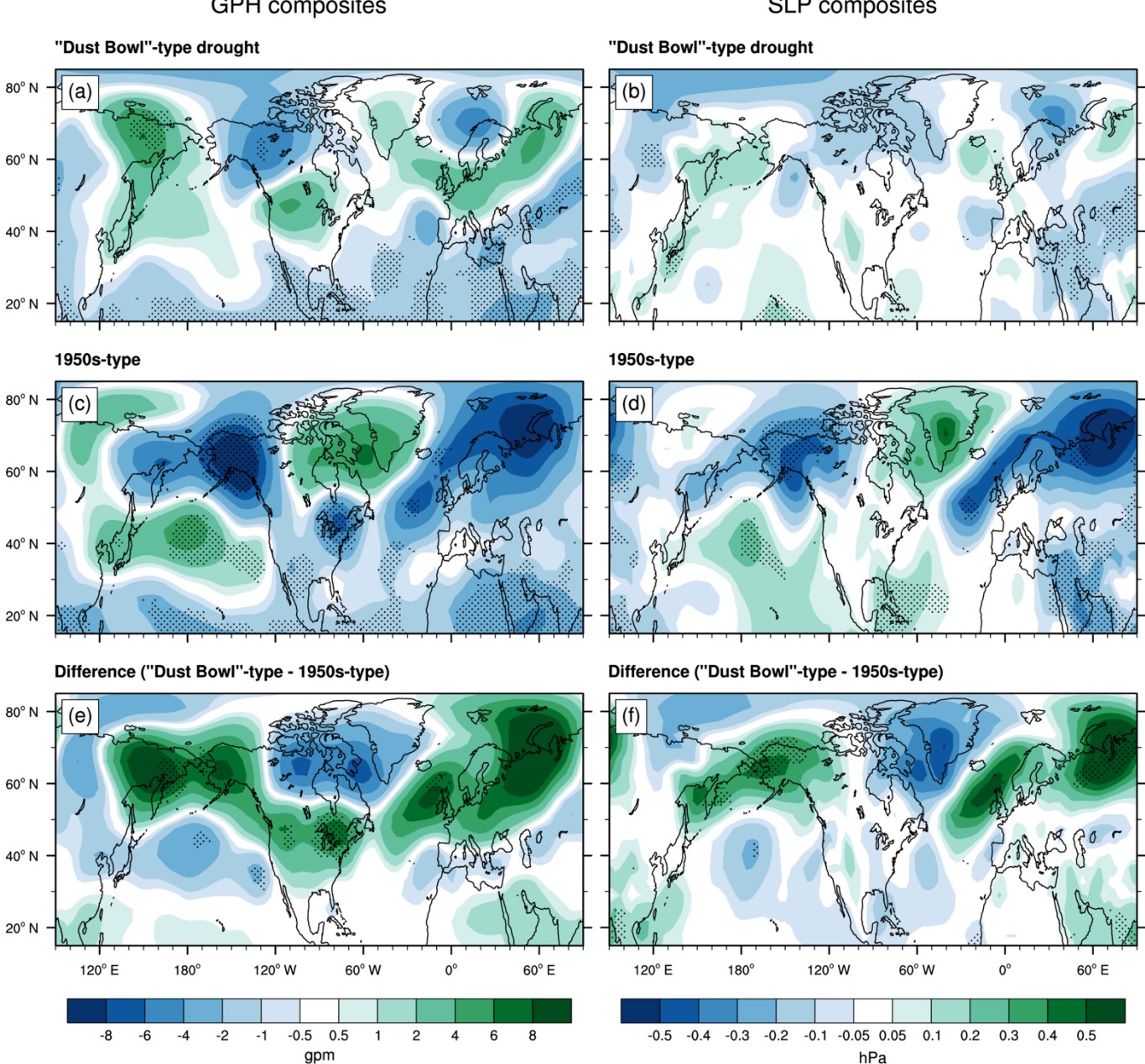

**Figure 2: Composites of 500 hPa geopotential height anomalies (left column) and sea-level pressure anomalies (right column) from EKF400 for "Dust Bowl"-type droughts (top row), 1950s-type droughts (middle row) and their difference (bottom row). Stippling indicates significance (p<0.05) based on a non-parametric Wilcoxon-Mann-Whitney test.**

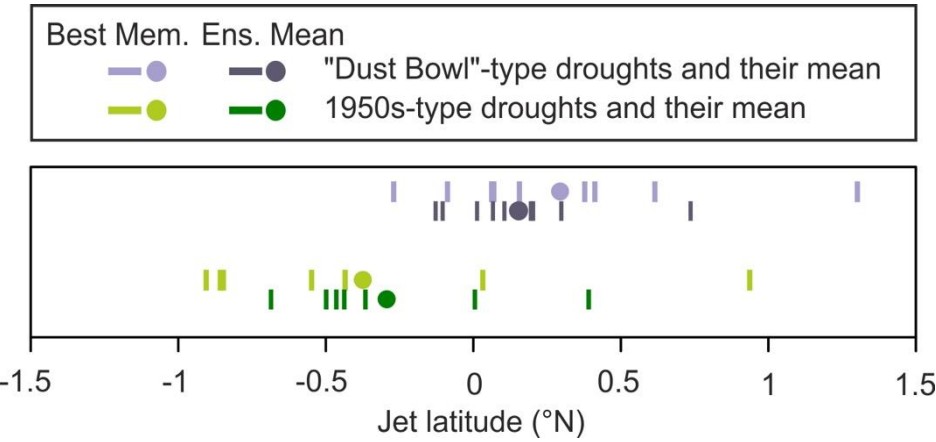

**Figure 3: Changes in the position of the subtropical jet over North America for "Dust Bowl"- type droughts (purple) and 1950s-type droughts (green). Anomalies are relative to the preceding and following 5-year period of the drought. Lines indicate individual drought periods, the circle indicates the drought-type mean.**

## T2m composites

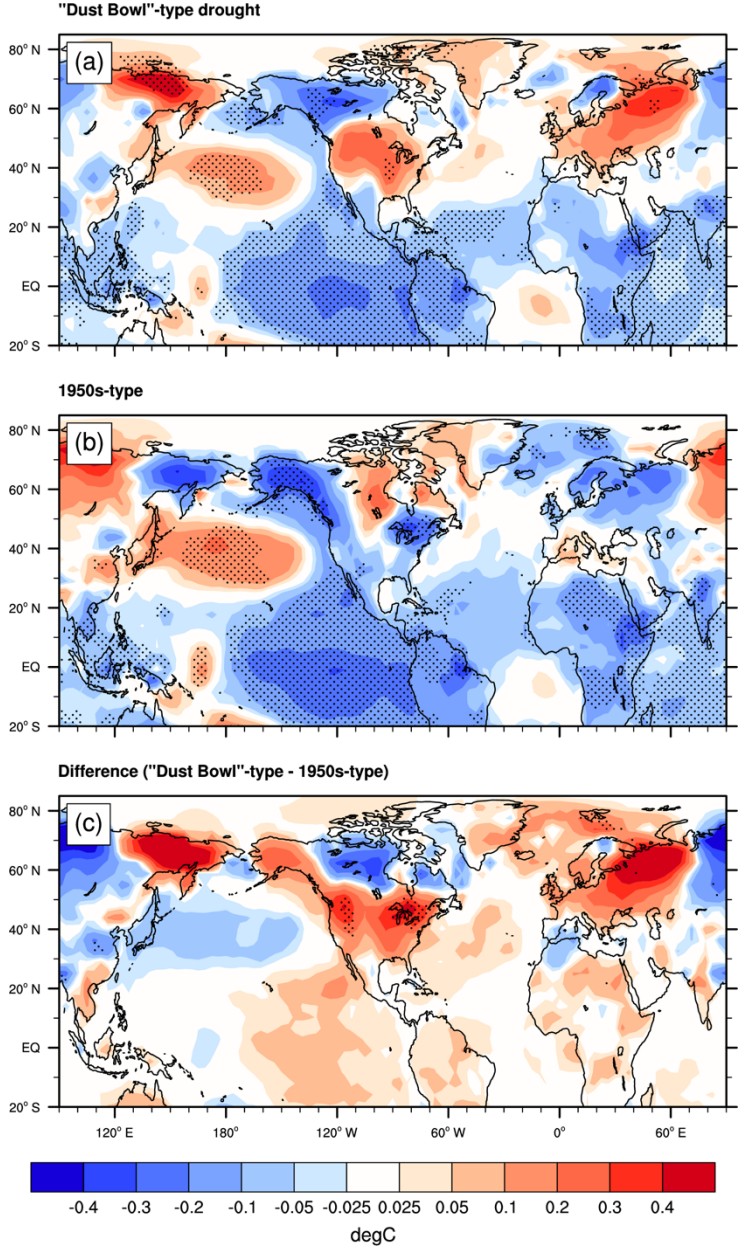

**Figure 4: Composites of 2-meter temperature from EKF400 for "Dust Bowl"-type droughts (a), 1950s-type droughts (b) and their difference (c). Stippling indicates significance (p<0.05) based on a non-parametric Wilcoxon-Mann-Whitney test.**

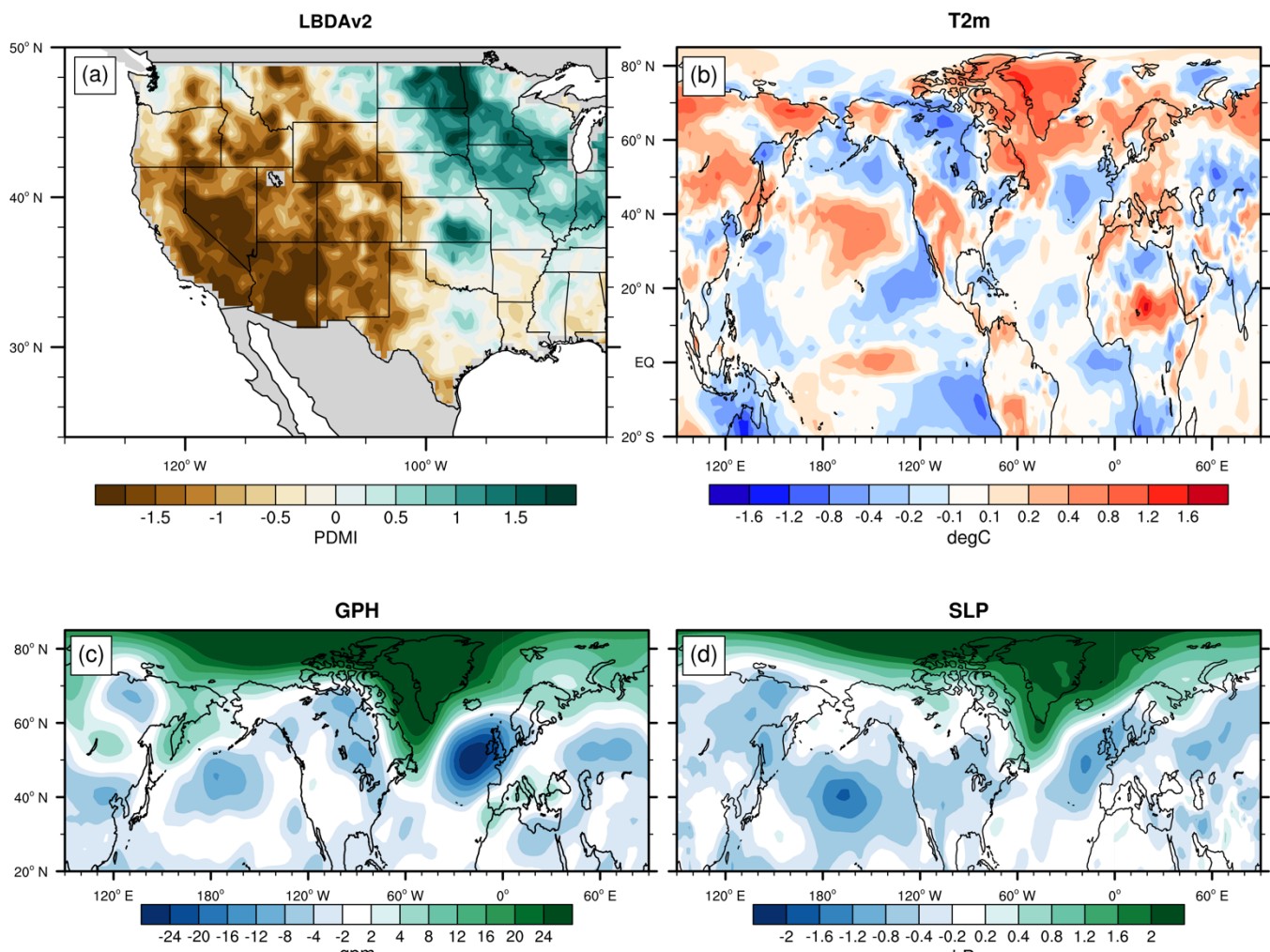

**Figure 5: The 21st century drought (2000-2015) in the PMDI of the LBDAv2 (a), composites relative to the preceding 5-year period and the following 3-year period (only 2016-2018 available) of 2-meter temperature (T2m) (b), 500 hPa geopotential height (GPH) (c) and sea-level pressure (SLP) (d) from the ERA-Interim reanalysis.**

## Tables

**Table 1:** Drought periods since 1600 based on clustering with LBDAv1, drought duration and attribution to cluster.

| # | LBDA Drought Periods | N $_{drought\ years}$ | Clustering |
|---|---|---|---|
| 17 | 2000 – 2005 | 5 | – |
| 16 | 1952 – 1965 | 14 | 1950s-type |
| 15 | 1931 – 1939 | 9 | "Dust Bowl"-type |
| 14 | 1892 – 1896 | 5 | 1950s-type |
| 13 | 1869 – 1874 | 6 | "Dust Bowl"-type |
| 12 | 1855 – 1866 | 12 | 1950s-type |
| 11 | 1817 – 1824 | 8 | 1950s-type |
| 10 | 1783– 1791 | 9 | 1950s-type |
| 9 | 1772– 1776 | 5 | "Dust Bowl"-type |
| 8 | 1753 – 1758 | 6 | "Dust Bowl"-type |
| 7 | 1734 – 1743 | 10 | 1950s-type |
| 6 | 1716 – 1720 | 5 | "Dust Bowl"-type |
| 5 | 1703 – 1710 | 8 | "Dust Bowl"-type |
| 4 | 1663 – 1671 | 9 | 1950s-type |
| 3 | 1652 – 1656 | 5 | "Dust Bowl"-type |
| 2 | 1644 – 1648 | 5 | "Dust Bowl"-type |
| 1 | 1624 – 1634 | 11 | "Dust Bowl"-type |