# Peer review of "Two types of North American droughts related to different atmospheric circulation patterns."

_Climate of the Past, 2019_

## Referee Comment (RC1) · Robert Jnglin Wills (Referee) · 21 Apr 2019

I find this to be an interesting paper and the conclusions can largely be supported by the work presented.

My only major concern is with the discussion of the links to the AMO. The AMO is generally associated with a center of action in the North Atlantic subpolar gyre (e.g., O'Reilly et al. 2016, Wills et al. 2019, and references therein), which shows no clear anomaly in either of the presented composites. To the extent that the Atlantic temperature anomalies are there at all, they (Fig. 4c) look more like NAO-coupled variability of the ocean gyre circulation (i.e., warming in the Gulf Stream and GIN seas, but cooling

in between; Curry and McCartney 2001, Eden and Jung 2001, Sun et al. 2015, Wills et al. 2019). Since these anomalies are weak anyway, I would limit your discussion of connections to the AMO, instead saying something like "there appear to be differences in Atlantic temperatures between the two drought types, and that this could be related to modes of Atlantic multi-decadal variability such as the AMO or the NAO-coupled variability of the gyre circulation, as discussed in the literature". Note that I've included a lot of Atlantic multidecadal variability literature here because of my own interest in that part of the story, and in case it is useful, but I don't actually think it is necessary to go into/reference all of it in this manuscript.

I have made a number of other scientific comments below that you should consider, and have pointed out some typos and wording problems. Overall this is a substantial contribution and the needed revisions are minor. Nice work.

Scientific questions/issues:

1. 17 droughts is a small number of degrees of freedom to be computing clusters from. Could you quantify what you mean by "most conclusive clustering result" or give some metric of how this clustering depends upon sampling? Furthermore, you then explain a principal component analysis based approach and this left me confused as to which method you were using. Are you using two separate methods to characterize the droughts? Do they get the same answer?

2. With regards to your methodology of "each drought period was first expressed relative to a reference period that comprised 5 years before and 5 years after the drought period", have you compared this to the simpler approach of using anomalies from the long-term mean? It seems that this would be a simple check and I would hope it doesn't make a huge difference.

3. Do you have an explanation why the SLP anomalies tend to be weaker / less significant than the GPH anomalies? Physically this would arise if the circulation anomalies were baroclinic (consistent with a shift of the subtropical jet in the longitude band of

[Figure]

Pacific/North America), but I am not sure the EKF400 reanalysis can be trusted to that great of a degree. Could it possible be reconstructing less of the SLP variance than it does the GPH variance? Are the differences in anomaly amplitude actually quantitatively different? It may be helpful to rescale the SLP colorbar and to consider my following comment.

4. Why do your GPH figures seem to have a mean over the plotted domain that is less than zero? This could be due to variability in the Southern Hemisphere that is not relevant here. Could you remove this so that the plots are easier to parse?

5. How is the 95% significance level computed for the figures? In particular, how are you computing the number of temporal degrees of freedom? It would be helpful to state this in the caption.

6. Please check that there are no major differences between a composite of SST and the T2M composite shown. No need to show it, but it would be good to check this and state whether there are any significant differences.

7. I don't fully agree with your interpretation of Fig. 4. There are not particularly stronger or more significant ocean T2M anomalies in the North Atlantic than the North Pacific. Given the larger influence of tropical SST anomalies on the atmospheric circulation (e.g., Kushnir et al. 2001), the different atmospheric anomalies are just as likely to result from the tropical Pacific or tropical Atlantic temperature anomalies, even though those anomalies are smaller and not significant. You state multiple times in the discussion that the warmer North Atlantic (while not significant) could explain this or that atmospheric change, but I don't think these results make a strong case for that, especially not for any role of the AMO, which should have larger-scale coherent warm anomalies focused in the subpolar gyre (such as those seen in Fig. 5). It may be helpful to consider Ruprich-Robert et al. 2017, which looks at the differing impacts between the tropical and extratropical component of "AMO" anomalies in a climate model.

8. Could you extend the latitude range of your T2M plot over the equator? Any SST

anomalies in the 0-20°S latitude range could still have a large impact on the atmospheric circulation in the Northern Hemisphere.

Technical corrections:

- Page 1, Line 14 typo: "show" should be "shows"

- Page 1, Line 16-17: positive and negative anomalies in what index?

- Page 2, Line 4-6: the words "most relevant" are not very precise, consider rephrasing

- Page 2, Line 11: Is "moisture interpretation" a vocabulary word I am not aware of, or is this simply a wording problem where you should have said "are mostly restricted to interpretations as temperature and moisture"?

- Page 2, Line 13: typo, extraneous "of" after behind

- Consider referencing Enfield et al. 2001 as well for the Atlantic SST influence on multi-decadal drought

- Page 3, Line 18 typo: "or" instead of "of"

- Your abstract had me wondering why only summer SST/SLP/GPH is relevant. If you say you are looking at summer drought, then it would become clear why, and you then don't even need to say that it is summer SST/SLP/GPH.

- Page 4, Line 6: please state how/why the ensemble members differ

- Page 4, Line 25-25: I think "opposed to decadal variability" is not the correct word choice for what you are saying. Should be "compared to decadal variability" instead.

- Page 6, Line 8: missing word(s) between La Niña and El Niño

- Page 6, Line 9/10: twice you say "three" where I think you mean "two"

- Page 6, Line 17 typo: "at in"

- Page 6, Line 17: former and latter are both singular, and you should use "exhibits"

with them, not "exhibit"

- May not Mai

- Mid-19th not mit-19th

- Page 8, Lines 19-20: the second half of this sentence needs to be reworded, this word order (especially with contribute at the end) does not work in English

- Page 8, Line 27: "turn-of-the-century drought" not "turn of the century"

- First sentence of conclusions: please add that this is the first time this has been studied in a climate reconstruction, because there have of course been model-based studies

References:

Curry, R. G., and M. S. McCartney, 2001: Ocean gyre circulation changes associated with the North Atlantic Oscillation. J. Phys. Oceanogr., 31, 3374–3400, https://doi.org/10.1175/1520-0485(2001)031,3374:OGCCAW.2.0.CO;2.

Eden, C., and T. Jung, 2001: North Atlantic interdecadal variability: Oceanic response to the North Atlantic Oscillation (1865–1997). J. Climate, 14, 676–691, https://doi.org/10.1175/1520-0442(2001)014,0676:NAIVOR.2.0.CO;2.

Kushnir, Y., W. A. Robinson, I. Blade', N. M. J. Hall, S. Peng, and R. Sutton, 2002: Atmospheric GCM response to extratropical SST anomalies: Synthesis and evaluation. J. Climate, 15, 2233–2256.

O'Reilly, C. H., M. Huber, T. Woollings, and L. Zanna, 2016: The signature of low-frequency oceanic forcing in the Atlantic multidecadal oscillation. Geophys. Res. Lett., 43, 2810–2818, https://doi.org/10.1002/2016GL067925

Ruprich-Robert, Y., R. Msadek, F. Castruccio, S. Yeager, T. Delworth, and G. Danabasoglu, 2017: Assessing the climate impacts of the observed Atlantic multidecadal

variability using the GFDL CM2. 1 and NCAR CESM1 global coupled models. J. Climate, 30, 2785–2810, https://doi.org/10.1175/JCLI-D-16-0127.1

Sun, C., J. Li, and F.-F. Jin, 2015: A delayed oscillator model for the quasi-periodic multidecadal variability of the NAO. Climate Dyn., 45, 2083–2099, https://doi.org/10.1007/s00382-014-2459-z.

Wills, R. C. J., Armour, K. C., Battisti, D. S., & Hartmann, D. L. (2019). Ocean-atmosphere dynamic coupling fundamental to the Atlantic Multidecadal Oscillation. Journal of Climate, 32(1), 251–272

---

## Referee Comment (RC2) · Anonymous Referee #2 · 22 Jun 2019

Review for Burgdorf et al., "Two types of North American droughts related to different atmospheric circulation patterns"

Burgdorf and colleagues, motivated to better understand drought forcing in North America, use the LBDA and EKF400 datasets to relate multiyear droughts in North America (via LBDA) to their synoptic circulation drivers (via EKF400) over a sufficiently long record to make robust claims.

The authors rely on clustering analysis of multiyear drought events (5-yr running mean on the standardized PDSI values) to identify their prevailing spatial patterns. They find two dominant modes of soil moisture anomalies (consistent with previous findings),

and building on work, are then positioned to link those patterns to their atmospheric drivers via the EKF400 data assimilation product. They find (generally consistent with the previous literature) that particular configurations of ocean-atmosphere variability select for different drought types.

Overall the paper appears to be in a position to make a nice contribution. I have a few larger comments and some minor ones the authors might find helpful in a revision.

Major comments:

1. How does the spatial domain presented (page 5, line 4) influence the clustering of the drought events and thus the spatial patterns presented? Presumably the clustering is quite sensitive to the domain selection and its odd that Figs. 1 and 5a use a constrained North American domain to show the drought patterns, while the rest of the analysis puts North America more fully in perspective. The LBDA v1 on which the central analysis is based, encompasses all of North America, so I wonder why the authors chose to constrain their analysis in such a way, particularly given their emphasis on both pattern identification (which is likely domain-sensitive) and synoptic scale circulation on such drought events. I recognize the authors' point at page 3, line 31 about version 2 being more limited spatially. But since the 20th C. drought is dropped from the central analysis, why not expand the domain to encompass all of NA, or at the least, all of CONUS (as in Fye et al.)?

2. I wonder about the comparison of EKF400 anomalies to the LBDA anomalies: do they share the same standardization intervals? I *believe* the LBDA is standardized relative to the instrumental period of 1931-90 (could be wrong here), and then the authors here take the 5-year running mean to define a drought event (plus a spatial scale threshold). The GPH analysis is just relative to +/- 5 years centered on the drought. I wonder if the atmospheric fields should first be centered to the same interval as the LBDA and then those EKF400 anomalies can be composited with the +/- 5 year approach. This may make some more consistent GPH, T2M, and SLP patterns

emerge.

3. Are the EKF400 T2M data representative of ocean skin temperatures? Certainly SSTs and T2M should share the same variations over climate timescales, but some kind of validation of that, or just using SSTs over ocean basins would be more sound for making claims about oceanic forcing.

4. It strikes me as a pretty large missed opportunity to not also leverage the PHYDA in this work as a check on EKF400 results, given the uncertainties in the latter that the authors concede (e.g., page 6, line 33) and the authors' search for robustness. As I understand it, EKF400 simulations are forced with SST reconstructions that use a number of the same proxies that are also then used in the data assimilation process itself, which seems potentially problematic. The authors' ability to make robust claims about wave trains, jet positions, and SSTs would be greatly enhanced if there is consistency across more than one atmospheric reconstruction, which is now publicly available.

5. Updating the Fye et al. paper seems to be a central motivation in this work and there are places where contrasts are drawn between the findings here and those in Fye, which is interesting, but it would be great to have the reasons for those differences explained or hypothesized about a bit more.

6. Finally, the outlier pattern associated with the most recent drought is really quite compelling as the authors suggest this one is anomalous based on their pattern clustering. Are there any droughts in the original two clusters (Dust Bowl and 1950s) that look somewhat like the modern drought? Some more validation of that finding would be really great. Could it be a methodological artifact due to its being in version 2 and not 1, and the need to put v2 (PMDI) on equal footing with v1 (PDSI)? It might be easier to drop this from the paper and do a more rigorous treatment of it in a separate analysis.

Minor comments:

1. P2, L27: You cite internal variability here; a recent Cook et al. 2018 paper ("Revisiting the Leading Drivers of Pacific Coastal Drought Variability in the Contiguous United States," Journal of Climate) shows that there are numerous ocean-atmosphere configurations that can give rise to the same drought pattern in the West Coast of North America.

2. P5, L5: is the PDSI < -1 consistent with Fye?

3. P5, L10: point to the Supplemental Figures here?

4. P6, L9: You discuss two drought types, but in this and the subsequent sentences you reference three.

5. Online it's fine, but in a print out, Fig. 2's color bar is difficult to discern.

6. P6, L29: Are these statistical tests on the patterns of droughts or the jet positions? As written it's not clear. (Seems like it should be on the jet positions.)

7. the quotes around "Dust Bowl" and such are upside-down(?); usage of e.g. requires a parenthetical, etc.; please just check the manuscript for the typos throughout.

---

## Editor Comment (EC1) · Alberto Reyes (Editor) · 1 Jul 2019

Dear Dr. Burgdorf and co-authors,

The discussion period for your manuscript is over, and two reviewers have posted comments. Both reviewers engaged thoughtfully with the manuscript, and I agree with their positive assessment of the work and its potential contribution to the field. The reviewers have several suggestions for additional consideration, some points of clarification, and technical corrections. My sense is that you can address the suggestions and concerns of the reviewers with relatively minor revision.

[Figure]

Please respond to the reviewer comments in the online Discussion forum. If you intend to make any changes to your manuscript in response to these comments, please clearly indicate the nature of these changes in your response. Once I have reviewed your response, I fully anticipate inviting you to submit a revised manuscript for further consideration.

Sincerely, Alberto Reyes

———————————————————

---

## Author Comment (AC1) · 21 Jul 2019

The comment was uploaded in the form of a supplement:
https://www.clim-past-discuss.net/cp-2019-22/cp-2019-22-AC1-supplement.pdf

---

## Author Comment (AC2) · 21 Jul 2019

**Reviewer #2 (Anonymous Referee)**

*Burgdorf and colleagues, motivated to better understand drought forcing in North America, use the LBDA and EKF400 datasets to relate multi-year droughts in North America (via LBDA) to their synoptic circulation drivers (via EKF400) over a sufficiently long record to make robust claims.*
*The authors rely on clustering analysis of multiyear drought events (5-yr running mean on the standardized PDSI values) to identify their prevailing spatial patterns. They find two dominant modes of soil moisture anomalies (consistent with previous findings), and building on work, are then positioned to link those patterns to their atmospheric drivers via the EKF400 data assimilation product. They find (generally consistent with the previous literature) that particular configurations of ocean-atmosphere variability select for different drought types.*
*Overall the paper appears to be in a position to make a nice contribution. I have a few larger comments and some minor ones the authors might find helpful in a revision.*

We thank the reviewer for the positive feedback and constructive comments and suggestions. In the following, we will take a stance on them.

**Major comments**
*1. How does the spatial domain presented (page 5, line 4) influence the clustering of the drought events and thus the spatial patterns presented? Presumably the clustering is quite sensitive to the domain selection and its odd that Figs. 1 and 5a use a constrained North American domain to show the drought patterns, while the rest of the analysis puts North America more fully in perspective. The LBDA v1 on which the central analysis is based, encompasses all of North America, so I wonder why the authors chose to constrain their analysis in such a way, particularly given their emphasis on both pattern identification (which is likely domain-sensitive) and synoptic scale circulation on such drought events. I recognize the authors' point at page 3, line 31 about version 2 being more limited spatially. But since the 20th C. drought is dropped from the central analysis, why not expand the domain to encompass all of NA, or at the least, all of CONUS (as in Fye et al.)?*

That is a legitimate point we focused on for a considerable amount of time. We first looked at the first version of the NADA (Cook et al., 2004, 2008) and compared the droughts resulting from our detection method with the droughts found by Fye et al. (2003) who used an earlier version of the PDSI (Cook et al., 1996, 1999). In these first examinations, we looked at the entire domain (all of North America) and ended up with a very similar set of droughts. We then looked at the droughts in the even more advanced version the LBDAv1 (Cook et al., 2010) which again showed an almost identical drought catalog, affirming the robustness of our approach. Since all multi-annual droughts we found mainly affect the Great Plains and the Southwest, we tested the idea of subsetting our domain to even better capture the spatial signal of droughts in this particular drought-prone area. In doing so, our detection method revealed in addition to the pervious drought catalog, further multi-annual drought periods that are very prominent in the relevant domain but remain disguised when looking at entire North America.
So while we argue that the development and subsistence of droughts in North America are driven by large scale circulation, we recognized that their spatial signature is more limited to particular regions in central and western North America which calls for a more regional vantage point in terms of the detection and classification of droughts. We ended up choosing a domain that includes prominent drought regions in the Mississippi Valley, Northern Mexico, and the southern Canadian Great Plains but excludes the East Coast and the tropical South as well as most of Canada.

*2. I wonder about the comparison of EKF400 anomalies to the LBDA anomalies: do they share the same standardization intervals? I \*believe\* the LBDA is standardized relative to the instrumental period of 1931-90 (could be wrong here), and then the authors here take the 5-year running mean to define a drought event (plus a spatial scale threshold). The GPH analysis is just relative to +/- 5 years centered on the drought. I wonder if the atmospheric fields should first be centered to the same interval as the LBDA and then those EKF400 anomalies can be composited with the +/- 5 year approach. This may make some more consistent GPH, T2M, and SLP patterns emerge.*

In our opinion this is negligible for the following two reasons: Firstly, we would like to point out that since we are looking at anomalies alone, the absolute values and how they are standardized are not relevant. Secondly, we do not directly compare LBDA and EKF400 anomalies. We use the LBDA solely to detect multi-annual droughts (± 5 year running mean plus a spatial scale threshold) based on a standardized PDSI relative to the instrumental period. The EKF400 which does not have the same standardization interval is analyzed strictly independently of the LBDA, we just apply the composite analysis over the detected drought years.

*3. Are the EKF400 T2M data representative of ocean skin temperatures? Certainly SSTs and T2M should share the same variations over climate timescales, but some kind of validation of that, or just using SSTs over ocean basins would be more sound for making claims about oceanic forcing.*

We are currently looking into this comparison and will provide some statement/validation in the revised version addressing this matter. Furthermore, we plan to include a new supplementary figure in the revised manuscript showing the comparison of SSTs and T2m.

*4. It strikes me as a pretty large missed opportunity to not also leverage the PHYDA in this work as a check on EKF400 results, given the uncertainties in the latter that the authors concede (e.g., page 6, line 33) and the authors' search for robustness. As I understand it, EKF400 simulations are forced with SST reconstructions that use a number of the same proxies that are also then used in the data assimilation process itself, which seems potentially problematic. The authors' ability to make robust claims about wave trains, jet positions, and SSTs would be greatly enhanced if there is consistency across more than one atmospheric reconstruction, which is now publicly available.*

We discussed before submitting our paper whether to include some comparison with PHYDA or not. Since Steiger et al. (2018) show that PHYDA is highly consistent with the LBDA we decided to stick with the LBDA which has a higher spatial resolution. Nevertheless, we will perform our T2m composite analysis with the PHYDA dataset to compare with our results and add some comparison to the revised manuscript. Unfortunately, T2m is the only variable we can compare with our EKF400 analyses, PHYDA does not include SLP and GPH fields.

*5. Updating the Fye et al. paper seems to be a central motivation in this work and there are places where contrasts are drawn between the findings here and those in Fye, which is interesting, but it would be great to have the reasons for those differences explained or hypothesized about a bit more.*

A possible hypothesis for the slightly different drought catalogue in Fye at al., 2003 and our study is the fact that they 1. use a different, less sophisticated version of PDSI reconstruction, 2. use a different drought detection metric and 3. use the entire domain over the contiguous U.S. Given these methodological differences, it is rather remarkable how similar the results are, pointing to their robustness.

*6. Finally, the outlier pattern associated with the most recent drought is really quite compelling as the authors suggest this one is anomalous based on their pattern clustering. Are there any droughts in the original two clusters (Dust Bowl and 1950s) that look somewhat like the modern drought? Some more validation of that finding would be really great. Could it be a methodological artifact due to its being in*

*version 2 and not 1, and the need to put v2 (PMDI) on equal footing with v1 (PDSI)? It might be easier to drop this from the paper and do a more rigorous treatment of it in a separate analysis.*

There is one drought (1652-1656) in our catalog of 16 droughts that somewhat resembles the turn-of-the-century drought. It also features the unique (among the other droughts in our catalog) spatial pattern with a diagonal divide between dry anomalies along the entire Westcoast and Southwest and wet anomalies (!) in the north-eastern Great Plain stretching down south along the Mississippi Valley. We will add plots of all the individual multi-annual droughts as a new supplementary figure so droughts can be looked at individually. We also add here the correlations plot showing how different the turn-of-the-century drought is compared to the two drought types. We don't think that the fact that the drought has a very distinct spatial signature is an artifact caused by the calibration of the LBDAv2 to fit the LBDAv1. The differences are only minor and would not explain the very different spatial pattern featuring the 21th-century drought.

[Figure]

**Figure1** | Correlations (pearson) between the turn-of-the-century drought and the "Dust Bowl"-type drought **A** and between the turn-of-the-century drought and the 1950s-type drought respectively **B.** Each point depicts a gridbox.

**Minor comments**
- *P2, L27: You cite internal variability here; a recent Cook et al. 2018 paper ("Revisiting the Leading Drivers of Pacific Coastal Drought Variability in the Contiguous United States," Journal of Climate) shows that there are numerous ocean-atmosphere configurations that can give rise to the same drought pattern in the West Coast of North America.*
  Thank you, we will add that reference to the revised version and point out the combined influence of atmospheric variability and forced ocean low-frequency variability.

- *P5, L5: is the PDSI < -1 consistent with Fye?*
  Yes. However, Fye at all used a different approach for drought detection (they are identifying decadal moisture regimes).

- *P5, L10: point to the Supplemental Figures here?*
  Thank you, that is a good idea, we will point to the supplement here.

- *P6, L9: You discuss two drought types, but in this and the subsequent sentences you reference three.*
  That is a mistake and will be corrected in the revised version.

- *Online it's fine, but in a print out, Fig. 2's color bar is difficult to discern.*
  Thank you for pointing that out, we will adjust the color bar in the revised version.

- *P6, L29: Are these statistical tests on the patterns of droughts or the jet positions? As written it's not clear. (Seems like it should be on the jet positions.)*
  They are on the jet position. We will formulate this more clearly in the revised version.

- *the quotes around "Dust Bowl" and such are upside-down(?); usage of e.g. requires a parenthetical, etc.; please just check the manuscript for the typos throughout.*
  Thank you for pointing out these typos, we will correct them and check for further typos.

---

## Author Response (AR1)

**Reviewer #1 (Robert Jnglin Wills)**

*I find this to be an interesting paper and the conclusions can largely be supported by the work presented. Overall this is a substantial contribution and the needed revisions are minor. Nice work.*

We thank the reviewer for the careful revision of the manuscript and appreciate the positive feedback and the helpful comments and suggestions.

*My only major concern is with the discussion of the links to the AMO. The AMO is generally associated with a center of action in the North Atlantic subpolar gyre (e.g., O'Reilly et al. 2016, Wills et al. 2019, and references therein), which shows no clear anomaly in either of the resented composites. To the extent that the Atlantic temperature anomalies are there at all, they (Fig. 4c) look more like NAO-coupled variability of the ocean gyre circulation (i.e., warming in the Gulf Stream and GIN seas, but cooling in between; Curry and McCartney 2001, Eden and Jung 2001, Sun et al. 2015, Wills et al. 2019). Since these anomalies are weak anyway, I would limit your discussion of connections to the AMO, instead saying something like "there appear to be differences in Atlantic temperatures between the two drought types, and that this could be related to modes of Atlantic multi-decadal variability such as the AMO or the NAO-coupled variability of the gyre circulation, as discussed in the literature". Note that I've included a lot of Atlantic multidecadal variability literature here because of my own interest in that part of the story, and in case it is useful, but I don't actually think it is necessary to go into/reference all of it in this manuscript.*

We agree with the reviewer and have included the suggested phrase in our revised manuscript:
*"There appear to be differences in Atlantic temperatures between the two drought types (Fig. 4), which could be related to modes of Atlantic multi-decadal variability such as the AMO or the NAO-coupled variability of the gyre circulation as discussed in the literature (i.e., warming in the Gulf Stream and Greenland, Iceland and Norwegian Seas (GIN) seas, but cooling in between; Curry and McCartney, 2001; Eden and Jung, 2001; Sun et al., 2015; Wills et al., 2019)."*

**Scientific questions/issues:**

*17 droughts is a small number of degrees of freedom to be computing clusters from. Could you quantify what you mean by "most conclusive clustering result" or give some metric of how this clustering depends upon sampling? Furthermore, you then explain a principal component analysis based approach and this left me confused as to which method you were using. Are you using two separate methods to characterize the droughts? Do they get the same answer?*

A sample size of 17 droughts is admittedly small and quite sensitive to the number of clusters or generally the sampled area. We thus tried many different settings of the clustering by changing the cluster numbers, the chosen spatial and temporal domain as well as the clustering approach itself. We found that limiting the clustering to two instead of three clusters resulted in the more evident classification of the 17 droughts and is furthermore consistent with the literature (*e.g.*, Fye et al., 2003). In that process, we also decided to exclude the turn-of-the-century drought from the clustering because of its inherently different spatial signal compared to the other 16 droughts in our sample.
In terms of the clustering method: We tested two different clustering approaches: k-means clustering and ward clustering, both of which have their strengths and weaknesses. While both methods resulted in an almost identical classification of the droughts, they disagreed on the class affiliation of one drought. Therefore, we chose to combine the two clustering methods and make use of the method's strengths to more accurately define the right cluster for the remaining drought (page 6, l. 14-15): "Ward hierarchical clustering was used to determine the cluster centers, which were then used as a starting point for the

k-means clustering."). The combined approach of ward and k-means results in the identical drought affiliation as the ward method (Supplementary Figure S3 and S4), pointing to the robustness of the clustering.

In the revised version we mention that clustering is sensitive to the cluster numbers, the chosen spatial and temporal domain as well as the clustering approach itself. We note that limiting the cluster number to two results in a robust classification that is furthermore consistent with Fye et al. (2003).

*With regards to your methodology of "each drought period was first expressed relative to a reference period that comprised 5 years before and 5 years after the drought period", have you compared this to the simpler approach of using anomalies from the long-term mean? It seems that this would be a simple check and I would hope it doesn't make a huge difference.*

This is a good point, thank you. We did compare the ±5 years composites approach to other approaches for the case of SLP. Figure 1 shows the comparison of the drought type composites and differences between our ±5yrs approach (a,c,e) and anomalies from the long-term mean over the entire period (1604-2003) (b,d,f). Both approaches show qualitatively similar patterns and point towards their robustness. We added a sentence on that to the revised paper and the figure to the Supplementary material. However, using a common climatology is not a good option for variables that have strong centennial trends such as T2m and GPH. For these variables, spurious signals may appear as our drought sample is small and the two types of droughts are not equally distributed over time, so they will be aliased by global warming trends. A long term trend would have to be subtracted, raising numerous other questions as to what trend should be chosen and how it should be fitted. We believe that the ±5yr composite approach is much more suitable to capture the differences between the drought and non-drought periods in a satisfactory manner.

[Figure]

**Figure 1 |** SLP composites relative to the 1604-2003 long-term mean **(a,c,e)** and the composites relative to the 5yrs periods prior and after the drought **(b,d,f)**. The top row depicts the SLP composites of the "Dust Bowl"-type droughts, the middle row the 1950s-type droughts and the bottom row the difference between the two drought types.

*Do you have an explanation why the SLP anomalies tend to be weaker / less significant than the GPH anomalies? Physically this would arise if the circulation anomalies were baroclinic (consistent with a shift of the subtropical jet in the longitude band of Pacific/North America), but I am not sure the EKF400 reanalysis can be trusted to that great of a degree. Could it possible be reconstructing less of the SLP variance than it does the GPH variance? Are the differences in anomaly amplitude actually quantitatively different? It may be helpful to rescale the SLP colorbar and to consider my following comment.*

That is an interesting comment. We suspect that this might be due to the fact that the 500hPa field includes more of the temperature signal and thus performs statistically slightly better.
In the revised manuscript we improved the GPH and SLP colorbars to make the differences in the anomaly amplitude better visible.

*Why do your GPH figures seem to have a mean over the plotted domain that is less than zero? This could be due to variability in the Southern Hemisphere that is not relevant here. Could you remove this so that the plots are easier to parse?*

Well spotted. N-S asymmetries can play a role, but one has to keep in mind, that our approach is not mass conserving (applicable for SLP).

*How is the 95% significance level computed for the figures? In particular, how are you computing the number of temporal degrees of freedom? It would be helpful to state this in the caption.*

This is mentioned in the method section but we added it to the captions as well. All significant tests are based on a non-parametric Wilcoxon-Mann-Whitney test.

*Please check that there are no major differences between a composite of SST and the T2M composite shown. No need to show it, but it would be good to check this and state whether there are any significant differences.*

We compared the T2m fields with the SST fields that were used as input variables for the model. Figure 2 shows the comparison. It can be seen that over the ocean, there exists very little difference between the T2m (a-c) and the SST from the model input (d-f). We therefore conclude that the use of T2m is justified also over the ocean. We included at statement thereof in the revised manuscript.

[Figure]

**Figure 2 |** Composites of T2m **(a-c)** and the SSTs that were used as model input **(d-f)**. The top row depicts the temperature composites of the "Dust Bowl"-type droughts, middle row the 1950s-type droughts and the bottom row the difference between the two drought types.

*I don't fully agree with your interpretation of Fig. 4. There are not particularly stronger or more significant ocean T2M anomalies in the North Atlantic than the North Pacific. Given the larger influence of tropical SST anomalies on the atmospheric circulation (e.g., Kushnir et al. 2001), the different atmospheric anomalies are just as likely to result from the tropical Pacific or tropical Atlantic temperature anomalies,*

*even though those anomalies are smaller and not significant. You state multiple times in the discussion that the warmer North Atlantic (while not significant) could explain this or that atmospheric change, but I don't think these results make a strong case for that, especially not for any role of the AMO, which should have larger-scale coherent warm anomalies focused in the subpolar gyre (such as those seen in Fig. 5). It may be helpful to consider Ruprich-Robert et al. 2017, which looks at the differing impacts between the tropical and extratropical component of "AMO" anomalies in a climate model.*

In the revised version we limited our discussion on the connections between droughts and the AMO. Instead we point towards the differences in Atlantic temperatures between the two drought types that are potentially related to different modes of Atlantic multi-decadal variability.

*Could you extend the latitude range of your T2M plot over the equator? Any SST anomalies in the 0-20° S latitude range could still have a large impact on the atmospheric circulation in the Northern Hemisphere.*

We extended the T2m composite fields to 20°South in the revised manuscript to get a more comprehensive view.

**Technical corrections:**

Thank you for pointing out typos/wording problems. We corrected the following in the revised version.

- *Page 1, Line 14 typo: "show" should be "shows"*
- *Page 2, Line 13: typo, extraneous "of" after behind*
- *Page 3, Line 18 typo: "or" instead of "of"*
- *Page 4, Line 25-25: I think "opposed to decadal variability" is not the correct word choice for what you are saying. Should be "compared to decadal variability" instead.*
- *Page 6, Line 8: missing word(s) between La Niña and El Niño*
- *Page 6, Line 17 typo: "at in"*
- *May not Mai*
- *Mid-19th not mit-19th*
- *Page 6, Line 17: former and latter are both singular, and you should use "exhibits" with them, not "exhibit"*
- *Page 8, Line 27: "turn-of-the-century drought" not "turn of the century"*
- *Page 1, Line 16-17: positive and negative anomalies in what index?*

  We specified here: positive and negative **GPH** anomalies.

- *Page 2, Line 4-6: the words "most relevant" are not very precise, consider rephrasing*

  Agreed, we used "alarming" instead.

- *Page 2, Line 11: Is "moisture interpretation" a vocabulary word I am not aware of, or is this simply a wording problem where you should have said "are mostly restricted to interpretations as temperature and moisture"?*

  This is a wording problem, we used your suggestion instead.

- *Consider referencing Enfield et al. 2001 as well for the Atlantic SST influence on multi-decadal drought.*

  Good idea, we added Enfield et al. 2001 in the revised version.

- *Your abstract had me wondering why only summer SST/SLP/GPH is relevant. If you say you are looking at summer drought, then it would become clear why, and you then don't even need to say that it is summer SST/SLP/GPH.*

  Elegant, we added "summer" in the abstract to clarify that we are focussing on multi-annual droughts during summer.

- *Page 4, Line 6: please state how/why the ensemble members differ*

  We specified the sentence as follows in the revised manuscript to clarify how the 30 ensemble members in EKF400 differ:
  "EKF400 is a global, monthly, three-dimensional reconstruction based on an off-line assimilation of early instrumental data, documentary data, and proxies (tree ring width, late wood density) into an initial condition ensemble of 30 global model simulations using an Ensemble Kalman Filter technique. "

- *Page 6, Line 9/10: twice you say "three" where I think you mean "two"*

  Yes, that is a mistake, we corrected this to "two" in the revised version.

- *Page 8, Lines 19-20: the second half of this sentence needs to be reworded, this word order (especially with contribute at the end) does not work in English.*

  Agreed, we reworded this sentence in the revised manuscript to "Again, both Atlantic and Pacific might contribute. In particular, a negative PDO/La Niña like pattern over the Pacific has been shown to conduce to tropical expansion (Allen et al., 2014). Moreover, Atlantic SSTs were demonstrated to play a role in the form of a positive (negative) AMO for a poleward (equatorward) shift of the jet (Brönnimann et al., 2015)."

- *First sentence of conclusions: please add that this is the first time this has been studied in a climate reconstruction, because there have of course been model-based studies*

  Yes, that's a good point. We added this specification in our revised conclusion.

**Reviewer #2 (Anonymous Referee)**

*Burgdorf and colleagues, motivated to better understand drought forcing in North America, use the LBDA and EKF400 datasets to relate multi-year droughts in North America (via LBDA) to their synoptic circulation drivers (via EKF400) over a sufficiently long record to make robust claims.*

*The authors rely on clustering analysis of multiyear drought events (5-yr running mean on the standardized PDSI values) to identify their prevailing spatial patterns. They find two dominant modes of soil moisture anomalies (consistent with previous findings), and building on work, are then positioned to link those patterns to their atmospheric drivers via the EKF400 data assimilation product. They find (generally consistent with the previous literature) that particular configurations of ocean-atmosphere variability select for different drought types.*

*Overall the paper appears to be in a position to make a nice contribution. I have a few larger comments and some minor ones the authors might find helpful in a revision.*

We thank the reviewer for the positive feedback and constructive comments and suggestions. In the following, we will take a stance on them.

**Major comments**

*1. How does the spatial domain presented (page 5, line 4) influence the clustering of the drought events and thus the spatial patterns presented? Presumably the clustering is quite sensitive to the domain selection and its odd that Figs. 1 and 5a use a constrained North American domain to show the drought patterns, while the rest of the analysis puts North America more fully in perspective. The LBDA v1 on which the central analysis is based, encompasses all of North America, so I wonder why the authors chose to constrain their analysis in such a way, particularly given their emphasis on both pattern identification (which is likely domain-sensitive) and synoptic scale circulation on such drought events. I recognize the authors' point at page 3, line 31 about version 2 being more limited spatially. But since the 20th C. drought is dropped from the central analysis, why not expand the domain to encompass all of NA, or at the least, all of CONUS (as in Fye et al.)?*

That is a legitimate point we focused on for a considerable amount of time. We first looked at the first version of the NADA (Cook et al., 2004, 2008) and compared the droughts resulting from our detection method with the droughts found by Fye et al. (2003) who used an earlier version of the PDSI (Cook et al., 1996, 1999). In these first examinations, we looked at the entire domain (all of North America) and ended up with a very similar set of droughts. We then looked at the droughts in the even more advanced version the LBDAv1 (Cook et al., 2010) which again showed an almost identical drought catalog, affirming the robustness of our approach. Since all multi-annual droughts we found mainly affect the Great Plains and the Southwest, we tested the idea of subsetting our domain to even better capture the spatial signal of droughts in this particular drought-prone area. In doing so, our detection method revealed in addition to the pervious drought catalog, further multi-annual drought periods that are very prominent in the relevant domain but remain disguised when looking at entire North America.

So while we argue that the development and subsistence of droughts in North America are driven by large scale circulation, we recognized that their spatial signature is more limited to particular regions in central and western North America which calls for a more regional vantage point in terms of the detection and classification of droughts. We ended up choosing a domain that includes, beside the Great Plains and the Southwest, prominent drought regions in the Mississippi Valley, Northern Mexico, and the southern Canadian Great Plains. It however excludes the East Coast and the tropical South as well as Alaska and most of Canada.

We added an explanation as to why we chose that particular spatial domain for the detection of the multi-annual droughts.

*2. I wonder about the comparison of EKF400 anomalies to the LBDA anomalies: do they share the same standardization intervals? I \*believe\* the LBDA is standardized relative to the instrumental period of 1931-90 (could be wrong here), and then the authors here take the 5-year running mean to define a drought event (plus a spatial scale threshold). The GPH analysis is just relative to +/- 5 years centered on the drought. I wonder if the atmospheric fields should first be centered to the same interval as the LBDA and then those EKF400 anomalies can be composited with the +/- 5 year approach. This may make some more consistent GPH, T2M, and SLP patterns emerge.*

This suggestion, as we understand it, affects the results only insofar as standardisation is involved (centering to 1931-90 and then re-centering to +/-5 years cancels out), but this is a valid point. Standardisation emphasises the signal in the tropics and suppresses the high-latitude signal. Figure 1 with standardised anomalies (b,d,f) shows exactly that. However, to some extent this information is already included in the figure, namely in the form of the stippling of significant differences, which depends on the signal-to-noise ratio.

[Figure]

**Figure 1 |** GPH composites (±5yrs) based on our initial method (version A) **(a,c,e)** and GPH composites centered on the standardization interval (1931-90) prior to calculating the ±5yrs composites (version B) **(b,d,f)**. The top row depicts the GPH composites of the "Dust Bowl"-type droughts, the middle row the 1950s-type droughts and the bottom row the difference between the two drought types.

*3. Are the EKF400 T2M data representative of ocean skin temperatures? Certainly SSTs and T2M should share the same variations over climate timescales, but some kind of validation of that, or just using SSTs over ocean basins would be more sound for making claims about oceanic forcing.*

We compared the T2m fields with the SST fields that were used as input variables for the model. Figure 2 shows the comparison. It can be seen that over the ocean, there exists very little difference between the T2m (a-c) and the SSTs from the model input (d-f). We therefore conclude that the use of T2m is justified also over the ocean. We included a statement thereof in the revised manuscript.

[Figure]

**Figure 2 |** Composites of T2m **(a-c)** and the SSTs that were used as model input **(d-f)**. The top row depicts the temperature composites of the "Dust Bowl"-type droughts, middle row the 1950s-type droughts and the bottom row the difference between the two drought types.

*4. It strikes me as a pretty large missed opportunity to not also leverage the PHYDA in this work as a check on EKF400 results, given the uncertainties in the latter that the authors concede (e.g., page 6, line 33) and the authors' search for robustness. As I understand it, EKF400 simulations are forced with SST reconstructions that use a number of the same proxies that are also then used in the data assimilation process itself, which seems potentially problematic. The authors' ability to make robust claims about wave trains, jet positions, and SSTs would be greatly enhanced if there is consistency across more than one atmospheric reconstruction, which is now publicly available.*

Thank you for the suggestion. PHYDA does not allow interpretations of atmospheric circulation, but the reviewer is right that consistency across products should be checked. We therefore added a figure to the supplementary material (see Figure 3 below) that shows a comparison of temperature composites (±5yrs) of the latest four drought periods in the EKF400 (a-d), the PHYDA (e-h) and furthermore in BerkeleyEarth (i-l), 20CRv2c (m-p) and HadCRUT4.6 (q-t). It can be seen that there are both similarities and differences

among the different products. EKF400 is slightly closer to the observations over the landmasses than PHYDA. This can be expected since instrumental temperature data is assimilated. However, also PHYDA, that solely assimilates natural proxies (tree rings, corals, sclerosponges, ice cores, speleothems, lake sediments and marine sediment) corresponds well with the observations. For example over the Arctic and South East Asia PHYDA performs better than EKF400. PHYDA does assimilate data here whereas EKF400 does not. Both PHYDA and EKF400 as well as 20CRv2c have difficulties over tropical Africa and South America where very little proxies/inst. measurements are available and thus assimilated. The comparison of the two paleo reconstructions (EKF400, PHYDA) is a great opportunity for future work. However, within the scope of this paper we will focus on the EKF400 analysis since we have, besides T2m, atmospheric variables (GPH and SLP) available to analyse the atmospheric circulation patterns contributing/driving the different multi-annual droughts in the U.S.

[Figure]

[Figure]

**20CRv2c T2m droughts JJA**          **HadCRUT4.6 T2m droughts JJA**

**Figure 3 |** Temperature composites in five different datasets for the latest four droughts, the 1950s droughts (top row), the "Dust Bowl" drought (second row), the 1892-1996 drought (third row) and the 1869-1874 drought (bottom row). The T2m of the two palaeo reconstructions EKF400 **(a-d)** and PHYDA **(e-h)** are shown as well as the surface air temperature from the observational products Berkeley Earth **(i-l)** and HadCRUT6.4 **(q-t)** and additionally, the T2m from the 20CRv2c reanalysis **(m-p)**.

*5. Updating the Fye et al. paper seems to be a central motivation in this work and there are places where contrasts are drawn between the findings here and those in Fye, which is interesting, but it would be great to have the reasons for those differences explained or hypothesized about a bit more.*
A possible hypothesis for the slightly different drought catalogue in Fye at al., 2003 and our study is the fact that they 1. use a different, less sophisticated version of PDSI reconstruction, 2. use a different drought detection metric and 3. use the entire domain over the contiguous U.S. Given these methodological differences, it is rather remarkable how similar the results are, pointing to their robustness.

*6. Finally, the outlier pattern associated with the most recent drought is really quite compelling as the authors suggest this one is anomalous based on their pattern clustering. Are there any droughts in the original two clusters (Dust Bowl and 1950s) that look somewhat like the modern drought? Some more validation of that finding would be really great. Could it be a methodological artifact due to its being in version 2 and not 1, and the need to put v2 (PMDI) on equal footing with v1 (PDSI)? It might be easier to drop this from the paper and do a more rigorous treatment of it in a separate analysis.*

There is one drought (1652-1656) in our catalog of 16 droughts that somewhat resembles the turn-of-the-century drought. It also features the unique (among the other droughts in our catalog) spatial pattern with a diagonal divide between dry anomalies along the entire Westcoast and Southwest and wet anomalies (!) in the north-eastern Great Plain stretching down south along the Mississippi Valley. We added plots of all the individual multi-annual droughts as a new supplementary figure so droughts can be looked at individually. Figure 4 shows correlation plots of the turn-of-the-century drought vs. the "Dust Bowl"-type droughts (A) and the turn-of-the-century drought vs. 1950s-type droughts (B). It indicates how different the turn-of-the-century drought is compared to the two drought types. We don't think that the very distinct spatial signature of the turn-of-the-century drought is an artifact caused by the calibration of the LBDAv2 to fit the LBDAv1. Theses differences are only minor and would not explain the very different spatial pattern featuring the 21th-century drought.

[Figure]

**Figure 4 |** Correlations (pearson) between the turn-of-the-century drought and the "Dust Bowl"-type drought **A** and between the turn-of-the-century drought and the 1950s-type drought respectively **B.** Each point depicts a gridbox.

**Minor comments**
Thank you for the suggestions and for pointing out typos/wording problems. We corrected the following in the revised version.

- *P2, L27: You cite internal variability here; a recent Cook et al. 2018 paper ("Revisiting the Leading Drivers of Pacific Coastal Drought Variability in the Contiguous United States," Journal of Climate) shows that there are numerous ocean-atmosphere configurations that can give rise to the same drought pattern in the West Coast of North America.*
  Thank you, we added that reference to the revised manuscript and pointed out the combined influence of atmospheric variability and forced ocean low-frequency variability.

- *P5, L5: is the PDSI < -1 consistent with Fye?*
  Yes. However, Fye et al. used a different approach for drought detection (they are identifying decadal moisture regimes).

- *P5, L10: point to the Supplemental Figures here?*
  Thank you, that is a good idea, we did point to the supplement figure at this point in the revised version.

- *P6, L9: You discuss two drought types, but in this and the subsequent sentences you reference three.*
  That is a mistake that we corrected in the revised version.

- *Online it's fine, but in a print out, Fig. 2's color bar is difficult to discern.*
  Thank you for pointing that out, we adjusted the color bars of all figures in the revised version.

- *P6, L29: Are these statistical tests on the patterns of droughts or the jet positions? As written it's not clear. (Seems like it should be on the jet positions.)*
  They are on the jet position. We formulated this more clearly in the revised version.

- *the quotes around "Dust Bowl" and such are upside-down(?); usage of e.g. requires a parenthetical, etc.; please just check the manuscript for the typos throughout.*
  Thank you for pointing out these typos, we did correct them and checked for further typos in the revised manuscript.

[revised manuscript text omitted]

**Tables**

**Table 1:** Drought periods since 1600 based on clustering with LBDAv1, drought duration and attribution to cluster.

| # | LBDA Drought Periods | N $_{drought\ years}$ | Clustering |
|---|---|---|---|
| 17 | 2000 – 2005 | 5 | – |
| 16 | 1952 – 1965 | 14 | 1950s-type |
| 15 | 1931 – 1939 | 9 | "Dust Bowl"-type |
| 14 | 1892 – 1896 | 5 | 1950s-type |
| 13 | 1869 – 1874 | 6 | "Dust Bowl"-type |
| 12 | 1855 – 1866 | 12 | 1950s-type |
| 11 | 1817 – 1824 | 8 | 1950s-type |
| 10 | 1783– 1791 | 9 | 1950s-type |
| 9 | 1772– 1776 | 5 | "Dust Bowl"-type |
| 8 | 1753 – 1758 | 6 | "Dust Bowl"-type |
| 7 | 1734 – 1743 | 10 | 1950s-type |
| 6 | 1716 – 1720 | 5 | "Dust Bowl"-type |
| 5 | 1703 – 1710 | 8 | "Dust Bowl"-type |
| 4 | 1663 – 1671 | 9 | 1950s-type |
| 3 | 1652 – 1656 | 5 | "Dust Bowl"-type |
| 2 | 1644 – 1648 | 5 | "Dust Bowl"-type |
| 1 | 1624 – 1634 | 11 | "Dust Bowl"-type |